# Improved Estimation of End-Milling Parameters from Acoustic Emission Signals Using a Microphone Array Assisted by AI Modelling

**DOI:** 10.3390/s22103807

**Published:** 2022-05-17

**Authors:** Andrés Sio-Sever, Juan Manuel Lopez, César Asensio-Rivera, Antonio Vizan-Idoipe, Guillermo de Arcas

**Affiliations:** 1Grupo de Investigación en Instrumentación y Acústica Aplicada, Departamento de Ingeniería Mecánica, Universidad Politécnica de Madrid, 28040 Madrid, Spain; 2Grupo de Investigación en Instrumentación y Acústica Aplicada, Departamento de Telemática y Electrónica, Universidad Politécnica de Madrid, 28040 Madrid, Spain; juanmanuel.lopez@upm.es; 3Grupo de Investigación en Instrumentación y Acústica Aplicada, Departamento de Teoria de la Señal y Comunicaciones, Universidad Politécnica de Madrid, 28040 Madrid, Spain; c.asensio@upm.es; 4Departamento de Ingeniería Mecánica, Universidad Politécnica de Madrid, 28006 Madrid, Spain; antonio.vizan@upm.es

**Keywords:** acoustic emission, machining, milling, process monitoring, microphone, data-driven modelling, machine learning

## Abstract

This paper presents the implementation of a measurement system that uses a four microphone array and a data-driven algorithm to estimate depth of cut during end milling operations. The audible range acoustic emission signals captured with the microphones are combined using a spectral subtraction and a blind source separation algorithm to reduce the impact of noise and reverberation. Afterwards, a set of features are extracted from these signals which are finally fed into a nonlinear regression algorithm assisted by machine learning techniques for the contactless monitoring of the milling process. The main advantages of this algorithm lie in relatively simple implementation and good accuracy in its results, which reduce the variance of the current noncontact monitoring systems. To validate this method, the results have been compared with the values obtained with a precision dynamometer and a geometric model algorithm obtaining a mean error of 1% while maintaining an STD below 0.2 mm.

## 1. Introduction

Within the field of machining, process monitoring is becoming increasingly important due to trends that demand the manufacture of complex parts quickly and at low cost.

Classical methods for the study of machining processes are usually based on the use of dynamometers to study the forces applied on the workpiece during the cutting process [1,2,3]. This method can provide accurate information about the process, but the high cost of the sensors means that they are not widely used in industrial applications. In order to solve the cost problem, the use of ultrasonic [4] or electrostatic field sensors [5,6] for depth of cut and tool–workpiece contact measurements has been studied, but the delicate positioning of these systems makes it difficult to implement them in a way that does not interfere with the machine toolpath.

In this context, techniques based on the sensing of acoustic emission (AE) phenomena have gained increasing interest during the last few years due to their simplicity of use and the ability to monitor a wide variety of features, such as teeth breakage [7,8], runout [9,10] and chattering [11,12]. In particular, those techniques based on the use of noncontact sensors in the range of audible sound, such as different types of microphones [13,14], are particularly attractive due to the reduced intrusiveness, so they have been used for detecting the chattering phenomenon [15,16,17] and tool condition monitoring in conventional machining [18,19], high-speed machining [20] and abrasion processes [21].

The main problem with acoustic emission signals obtained by microphones is that the signal is heavily polluted by background noise and influenced by the acoustics of the environment. Generally speaking, the acquired signal includes information from the machining process, the rotation of machine bearings, the operation of various electric motors, waves reflected from metal surfaces and background noise generated by other machines and operators. 

Under these working conditions, the use of artificial intelligence (AI) techniques is presented as an interesting alternative to traditional signal processing methods to extract information of interest.

Among these solutions, neural network training methods with multiple layers are common choices to deal with complex processing [22,23]. Recently, more complex networks have been used to solve the tool state issue, such as convolutional networks combined with a wavelet transform [24,25]; or deep neural networks [26], which have a large number of hidden layers between the input and output layers. The main disadvantage of these systems lies in their high computational cost and the large amount of data that is necessary in their training to ensure accurate results free of overfitting [27].

Another method used for AI processing consists of classifier algorithms based on support vector machines (SVM), which represent signal features in different hyperplanes so that they can be easily separated [28,29]. It is necessary to take into account that this methodology provides results of discrete nature, which may not have the necessary resolution for the monitoring of continuous systems.

Finally, various machine learning techniques can be used to process acoustic data by logistic [30] or polynomial regression [31,32] for the application of tool condition monitoring and even for the prediction of cutting forces [33]. These algorithms allow both the use of features extracted by principal component analysis (PCA) and conventional signal features, which enables greater knowledge to be obtained regarding the variables present in the process.

Considering the previous state of the art, the aim of this paper is to develop a system for measuring the depth of cut $\left(a_{p}\right)$ by extracting features from the acoustic emission signal in the range of audible sound and implementing them in a nonlinear regression model, as it allows results to be obtained with higher resolution than classification models while maintaining a lower processing cost than neural networks [27].

In order to analyze the accuracy of the system, the results obtained will be compared with the results calculated using a dynamometer signal applied to a geometrical model of the system (Figure 1), which represents a highly accurate method for the calculation of the depth of cut [3].

Regarding the structure, Section 2 presents the hardware used for the measurement of the experimental process. Section 3 explains the signal processing method applied to the microphone signals. Section 4 shows the validation method based on the signal acquired by means of a precision dynamometer. In Section 5, the main results obtained are presented and analyzed qualitatively and quantitatively. Finally, Section 6 summarizes the main conclusions of this work.

## 2. Experimental Setup

This section describes the elements of the multichannel measurement system for the calculation of the depth of cut by nonintrusive methods.

Figure 2 shows the schematic of the system used for the experimentation, which consists of a computer numerical control (CNC) machining center, four microphones, a precision dynamometer and the elements associated with the acquisition and digitalization of the captured signals.

The experiments were performed on a DMG 1035 three-axis machining center with an 8 mm high-speed steel tool machining a 7075 aluminum piece (Figure 3). 

The measurement of acoustic phenomena inside the machining center is a complex problem, since it must be taken into account that it is a metallic structure that favors reverberation conditions in its interior, thus producing reflected waves of great magnitude.

In a previous study, it was found that acoustic emission waves picked up by microphones have mostly low frequency components [14]. Since the highest frequency component that would be measured was 200 Hz, then, its wavelength would be about 1720 mm. In turn, the width of the machine opening was 1140 mm which, being narrower than the wavelength of the signal, would work as a new emission point, as shown in Figure 2. 

The microphones used were four BSWA model MPA-201 precision microphones whose characteristics are presented in Table 1. They were calibrated using a BSWA CA111 calibrator to ensure identical sensitivity among the sensors, and arranged radially around the door to minimize the phase difference among them.

In parallel, the forces applied to the workpiece were measured by a Kistler 9257A dynamometer platform (Table 2) connected to a Kistler 5080 amplifier.

In both cases, the sensor signals were acquired by two NI-9234 data acquisition cards connected to a computer. Both cards work with a sampling frequency of 25.6 kHz with bipolar input, a dynamic range of 10 V and a resolution of 24 bits.

## 3. Data Processing

This section describes the processing of acoustic emission signals for estimating the depth of cut. Figure 4 shows a diagram of the process, which consists of: a two-stage preprocessing phase (1 and 2), feature extraction (3), elimination of outliers, and finally, the modeling of a nonlinear regression system assisted by machine learning (4).

### 3.1. Preprocessing

The first stage consists of the filtering of the signals acquired during the machining process in order to remove as much background noise as possible (Figure 4, points 1 and 2).

Firstly, it is necessary to eliminate the maximum amount of noise coming from the CNC machine itself, such as from the spindle, electric motors and servos.

This noise source has the advantage of being constant over time, so that it is particularly susceptible to being processed by spectral subtraction [34,35,36]. For this purpose, the sound produced by the machine performing the machining operation was acquired in the absence of the workpiece. Thus, a signal similar to the experimental one is obtained, but without the influence of the cutting sound. Finally, by performing a spectral subtraction, the resulting signals showed a lower incidence of the noisy elements of the machine (Figure 4, point 1).

The next step works with the background noise of the system. Due to the nature of the microphones, they pick up all acoustic signals that are present in the vicinity of the workspace, in adjacent rooms and even reflected waves from the source. In order to be able to study the machining signal with minimal incidence from other sources, an independent component analysis (ICA) blind source separation algorithm was applied [37] (Figure 4, point 2).

The first step in this process was to apply a whitening operation to the input data, so that any possible correlations were removed. Next, the marginal densities of the signal from each microphone were calculated, and then combined, to obtain the joint densities of the signals.

The next step was to apply an orthogonal transformation of the acquired signals in order to find the optimal rotation that maximized the non-normality of the marginal densities. This is an iterative process that will apply different rotations until the transformation that produces the maximum kurtosis value is found.

The assumption used in this case was that two combined signals were being acquired: the machining sound and the background noise [38]. Once the rotation that obtained the maximum kurtosis in an input signal was obtained, it was separated and considered as the background noise, picking up the signal with minimum kurtosis as the machining signal (Figure 5).

### 3.2. Feature Extraction

Starting from the machining signal obtained by signal separation (Figure 5), the signal was divided into intervals of 5 tool revolutions (between 0.15 and 0.25 s, depending on the spindle speed) by means of a Hann window [39], and each window was analyzed separately.

The experiments were conducted by changing the value of three different variables: depth of cut $\left(a_{p}\right)$, feed $\left(f_{z}\right)$ and spindle speed $\left(n\right)$, which require a different set of features to analyze their influence in the process (Figure 4, item 3). 

Under stable conditions of spindle speed and feed, the values of the depth of cut can be estimated by the following set of features:

First six harmonics of the FFT: the shape of the force signal given by the ideal geometrical model [3] has a frequency spectrum where there is energy only in the harmonics of the rotational speed of the tool [40]. By analyzing these peaks, the properties of the cutting process taking place can be calculated (Figure 6a). In this case, the first 6 harmonics were used because their peaks were consistently more than 10 dB away from the background noise level, which is not possible from the sixth harmonic onwards.

The 11th, 12th, 13th and 14th third of octave: from the sixth harmonic onwards, the magnitude of the frequency peaks was greatly reduced, being almost embedded in the background noise of the signal. To analyze these sections, the energy of the third octave band in which the peaks were located was studied, as their magnitude changed depending on the machining conditions. From the 14th third of octave, the frequency peaks were not usually visible above the background noise.

Spectral slope: calculates the slope of the energy drop at high frequencies [41], thus studying the influence of the different machining parameters on the high frequency components. 

Additionally, any change in the values of the spindle speed was represented in the frequency spectrum by shifting the location of the harmonics to higher frequencies when the speed increased, and to lower frequencies when the speed was reduced. These changes were dependent on the frequency value of the fundamental harmonic, whose position was determined by the rotation frequency of the spindle. The influence of this phenomenon can be estimated by the following frequency-focused features: spectral flux (Figure 6b), spectral roll-off point (75%), pitch and spectral centroid (Figure 6c) [41].

The amplitude of the resulting AE signal was highly dependent on the area of the metal chip being cut and the subsequent energy of the cut, which could be modeled by multiplying the depth of cut $\left(a_{p}\right)$ of the process by the feed value $\left(f_{z}\right)$. Since the estimation of the depth of cut had already been addressed, the resulting influence of different feed values was estimated by measuring the following features: root mean square (RMS) (Figure 6d), spectral flatness, spectral spread, Shannon entropy and spectral entropy [41].

#### 3.2.1. Feature Influence and Explainability

The selected features were analyzed with an F-test, which calculates the negative logarithm *p*-value of each feature and then represents those values as predictor importance scores, where higher values indicate the most representative features of the system (Figure 7).

These results indicate that the frequency peaks of the signal are the most important factor in estimating the depth of cut values of an unknown process, which is consistent with the initial hypothesis. Additionally, similar tests were performed where the objective was to differentiate the different feed and speed values, and whose results also supported the current feature selection. 

#### 3.2.2. Elimination of Outliers

The training of the regression model must be performed using data with stable and properly labeled features. For this reason, in each data set with the same machining conditions, the features whose values were beyond two standard deviations from the mean were removed, thus eliminating the influence of outliers in the regression model. 

### 3.3. Creation of the Regression Model

The following algorithm calculates the parameters of a nonlinear regression function assisted by machine learning (Figure 4, point 4). The algorithm used is an exponential regression model that uses a Gaussian process regression (GPR) for the calculation of parameters [42].

Assuming a training set $\left\{\left(x_{i},y_{i}\right);i=1,2,\ldots ,n\right\}$ where $x_{i}$ and $y_{i}$ are real numbers drawn from an unknown distribution, the Gaussian regression model predicts the value of a new output variable from a new input variable.

This regression system produces a model with a structure similar to:(1)$$y=x^{T}\beta +\varepsilon$$
where $\varepsilon \;∼\;N\left(0,\sigma^{2}\right)$ and β represent the coefficients estimated from the data.

The GPR explains the response of a system by introducing latent variables $f\left(x_{i}\right),\;i=1,2,\ldots ,n$ of a Gaussian process and explicit basis functions *h*. The covariance function of the latent variables captures the smoothness of the response and the basis functions project the input variables *x* into a *p*-dimensional feature space.

A Gaussian process (GP) is a set of random variables such that any finite number of them possess a joint Gaussian distribution. A Gaussian process is defined by its mean function mx and a covariance function kx,x′. This implies that, if fx,x∈ℝd is a Gaussian Process, then Efx=mx and Covfx,fx′=Efx−mxfx′−mx′=kx,x′.

Considering the model:(2)hxTβ+fx
where fx ∼ GP0,kx,x′, fx is composed of a zero-mean GP with a covariance function kx,x′, hx is a set of basis functions that transform the original feature vector x in ℝd into a new feature vector hx in ℝp and finally, β is a p-by-1 dimensional vector of the coefficients of the basis functions.

The above equation represents a GPR model, whose response *y* can be modeled as:(3)$$P\left(y_{i}|f\left(x_{i}\right),x_{i}\right)\;∼\;N\left(y_{i}|h{\left(x_{i}\right)}^{T}\beta +f\left(x_{i}\right),\sigma^{2}\right)$$

Therefore, a GPR model is a probabilistic model, where there is a latent variable $f\left(x_{i}\right)$ introduced for each observation $x_{i}$, which makes the regression model nonparametric.

In vector form, this model is equivalent to: $P\left(y|f,X\right)\;∼\;N\left(y|H\beta +f\left(x_{i}\right),\sigma^{2}\right)$
where:(4)$$X=\left(\begin{array}{c} x_{1}^{T}\\ x_{2}^{T}\\ \vdots \\ x_{n}^{T} \end{array}\right),$$
(5)$$y=\left(\begin{array}{c} y_{1}\\ y_{2}\\ \vdots \\ y_{n} \end{array}\right),\;$$
(6)$$H=\left(\begin{array}{c} h(x_{1}^{T})\\ h\left(x_{2}^{T}\right)\\ \vdots \\ h\left(x_{n}^{T}\right) \end{array}\right),$$
(7)$$f=\left(\begin{array}{c} f\left(x_{1}\right)\\ f(x_{2})\\ \vdots \\ f\left(x_{n}\right) \end{array}\right).$$

Then, the joint distribution of the latent variables $f\left(x_{1}\right),\;f\left(x_{2}\right),\;\ldots ,\;f\left(x_{n}\right)$ in the GPR model is defined as follows:(8)$$P\left(f|X\right)\;∼\;N\left(f|0,K\left(X,X\right)\right)\;$$

Finally, the results of the model are validated using the cross-validation method with 5 folds, thus ensuring that the resulting model does not overfit [43].

## 4. Validation 

Machining conditions in a milling process can be obtained in terms of the geometrical characteristics of the cutting tool [3]. This implies that the depth of cut $\left(a_{p}\right)$ can be related to the tool angles (entry angle $\left(\varphi_{en}\right)$, exit angle $\left(\varphi_{ex}\right)$, projected angle $\left(\varphi_{pr}\right)$ and helix angle $\left(\lambda_{s}\right)$) as shown in Figure 8a,b.

If a time base is available, the entry angle can be determined by measuring the time it takes the flute to complete that part of the rotation. This can be done using the force signal obtained with a dynamometer by adding a digital pulse (reference) for each revolution of the spindle as shown in Figure 8c. The associated time parameter is denoted as the entry time $\left(t_{en_{i}}\right)$ and it represents the time, measured from the reference pulse, that takes the cutting edge of the tool to come into contact with the workpiece on each spindle revolution. Thus, the entry time and entry angle are related by the spindle period $\left(T\right)$ as shown in Equation (9).
(9)$$t_{en}{}_{i}=\frac{\varphi_{en_{i}}}{2\pi }T$$

The projection time of the cutting tooth $\left(t_{pr}\right)$, depends on the projection angle of the active cutting tooth on the work plane of the workpiece being machined $\left(\varphi_{pr}\right)$, as described in Equation (10).
(10)$$t_{pr}{}_{i}=\frac{T}{2\pi }\varphi_{pr}{}_{i}=\frac{Ta_{p_{i}}tan\lambda_{s}}{\pi D}$$

Then, the depth of cut can be obtained in terms of the projection angle ($\varphi_{pr}$) and the rotation period of the tool according to Equation (11).
(11)$$a_{p}{}_{i}=\frac{\pi D}{Ttan\lambda_{s}}t_{pr}{}_{i}$$

These equations can be used to determine the value of the depth of cut with high accuracy if the timing parameters of the force signals can be obtained [3]. Therefore, this method has been used to validate the results of the developed system as shown in Figure 1. 

A Kistler 9257A dynamometer with a synchronized data acquisition board was used in parallel to the developed system to obtain the force signals during each machining process. The force signals were acquired simultaneously and the timing parameters were obtained as described in [3] to produce a reference value of the depth of cut. The values obtained with the developed and reference systems were then compared as described in the next section.

## 5. Results and Discussion

In the present work, different tests were carried out by varying the depth of cut, the feed rate and the rotational speed. From these tests, signals were obtained by means of four microphones and a dynamometer, which were processed by two different algorithms: one based on a regression model assisted by machine learning (Section 3) and the other based on the analysis of the geometric cutting model (Section 4).

### 5.1. Experimentation

The different tests were carried out under the conditions described in Table 3, varying the depth of cut between 1 and 10 mm in one-millimeter intervals. In turn, the values of feed rate and rotational speed were varied individually to check the behavior of the system in each situation, varying between different values of feed rate while maintaining a constant rotational speed (1200 rpm) and also varying between different values of rotational speed while maintaining a constant feed rate (0.08 mm).

To ensure consistency of results, each combination of the selected parameters was machined and recorded five times consecutively. Each of these operations had a duration of at least 5 s, thus producing a minimum of 25 s of machining signals for each combination of machining parameters. These conditions ensured that once the tests had been carried out, more than 30,000 individual cuts executed under different working conditions were available for analysis.

### 5.2. Calculation of the Depth of Cut with the Regression Model

The main results of the regression model were analyzed in terms of the root mean-square error (RMSE) for the different conditions of spindle speed and feed value (Figure 9). The model reported good accuracy in terms of absolute error, showing an average RMSE always below 0.2 mm, which represents a relative error of 1% on average. The RMSE increased with an increased spindle speed value, while decreasing with an increased feed value, which can be explained in both cases by the additional intensity of the acoustic emission resulting from having to cut a greater amount of material in each revolution, thus increasing the signal-to-noise ratio of the signal and making it easier for the model to recognize.

When the individual depths of cut were considered (Figure 10), it was noted that when the depth of cut was close to 1 mm, the system lost remarkable accuracy due to the reduction of the intensity of the acoustic emission, since it made it difficult to separate it from the rest of the noise present, having a signal-to-noise ratio of only 3 dB.

Additionally, the dispersion of the results was also measured to analyze the influence of the machining conditions on the standard mean deviation value (STD) (Figure 11). The dispersion of the results was reduced when the value of the feed parameter was increased, which, as in the previous case, can be explained due to the additional intensity of the acoustic emission resulting from having to cut a greater amount of material in each revolution.

Likewise, when the rotational speed of the tool was increased, there seemed to be a general increase of the dispersion values due to the reduction of the chip thickness cut in each revolution, but this behavior was not uniform in all cases.

Similarly, there was no uniformity regarding the differences in dispersion for different values of depth of cut (Figure 10 and Table 4).

### 5.3. Accuracy of the Results Obtained by the Dynamometer and the Geometric Model

The dynamometer values showed a small dispersion and a mean depth of cut value similar to the programmed value during the cutting operation. The absolute accuracy of the system remained stable regardless of feed rate, rotational speed or depth of cut (Figure 9 and Figure 11).

It can be seen that the results of the dynamometer slightly increased their dispersion as the chip thickness increased, contrary to the case of the microphones (Table 4). This is explained by the fact that the dynamometer does not have the background noise problems of the microphone, so that the increase in dispersion is due to the increase in uncertainty of the mathematical method used to calculate the depth of cut value [3].

The small error between the values obtained with the dynamometer and the microphones supports the hypothesis of using this system for noncontact monitoring of machining processes.

### 5.4. Comparison of Results

In general terms, the average values provided by the microphones were acceptable for a monitoring process, but did not achieve the accuracy of the direct contact measurement method. When machine learning is applied to the microphone signal, a dependence between the RMSE and the thickness of the cut chip could be observed (Figure 9), with the error increasing as the cutting speed increased or the feed rate was reduced.

In terms of dispersion, similar conclusions can be drawn, but showing a greater dependence on cutting speed (Figure 11).

The results provided by the dynamometer using the geometric cutting model show a high accuracy and a reduced dispersion, besides having the ability to work in different working conditions without the need for additional calibrations. On the other hand, the regression model used with the microphones is limited to work within the parameters with which it has been trained, being necessary to retrain with new values to be able to work outside those limits.

## 6. Conclusions

This paper presents the prototype of a system for measuring the depth of cut during a milling operation by analyzing the acoustic emission collected by four microphones and processed by a nonlinear regression method assisted by machine learning. The main ad-vantage of such a system is its reduced intrusiveness compared to its direct-contact alter-natives, which are affixed to the workpiece itself.

In a previous work, a single microphone implementation was developed based on traditional signal processing and a geometric model of the process [14]. This method required a simpler setup and calculation algorithm, but the signal was highly contaminated by other sources of noise and reverberation, limiting the system’s precision.

That setup produced results with an average relative error of 10%, which allowed the detection of the main changes in the milling conditions. In this paper, the average error was reduced from 10% to 1%, using blind source separation techniques on a four-element array and a data-driven nonlinear regression model obtained from a set of predefined features which had been derived from the underlying physics of the process, allowing more precise monitoring.

The current system has a limitation for very low depth of cut (e.g., 1 mm) due to a strong reduction in the signal to noise ratio. Under such circumstances, the amount of material being removed produces a very low intensity acoustic emission while the noise level is almost the same, so the system performance degrades significantly. Self-adapting variable-gain amplifiers could be used to improve performance for such situations, if needed.

In both cases, the systems were validated by comparing the results with those obtained using a precision dynamometer and an estimation algorithm based on the geometric model. 

Regarding the explainability, the geometric model directly relates different parameters of the milling tool and the machining process with the estimated value of depth. In contrast, the new data-driven model obtains the value of depth by analyzing the values of the frequency peaks, while accounting for feed changes with amplitude features and spindle speed changes with frequency features. This approach uses features selected from the current knowledge of the phenomena taking place in the process, which results in a more interpretable system compared to black box models such as deep neural networks.

The absolute accuracy of the system remains stable regardless of the values of feed rate, rotation speed or depth of cut. 

These results suggest the possibility of using noncontact sensors for the monitoring of machining processes, since they can provide reasonably accurate results even if they do not reach the precision values of modern dynamometers. These qualities are of potential interest in the large-scale manufacturing industry, as they allow the detection of defects in the machining process without exposing the sensors to excessively hostile environments, thus reducing the manufacture of defective parts and extending the lifetime of the sensors.

## Figures and Tables

**Figure 1 sensors-22-03807-f001:**
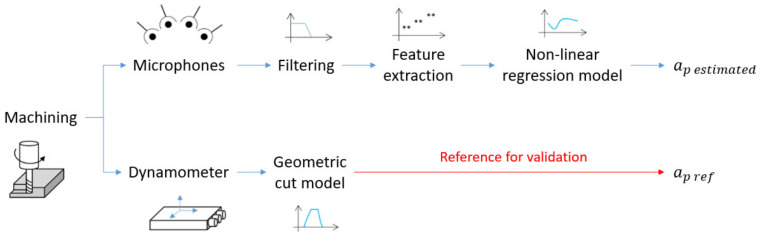
Schematic diagram of the experimental setup for the calculation of the depth of cut $\left(a_{p}\right)$.

**Figure 2 sensors-22-03807-f002:**
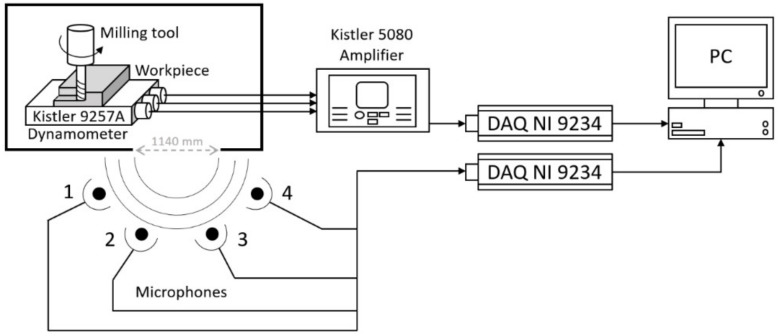
Diagram of the elements of the experiment.

**Figure 3 sensors-22-03807-f003:**
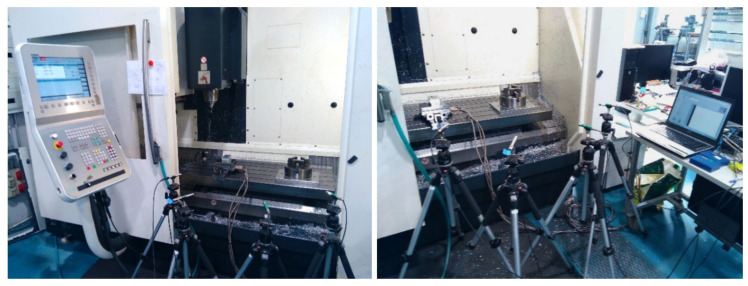
Pictures of the experimental setup.

**Figure 4 sensors-22-03807-f004:**
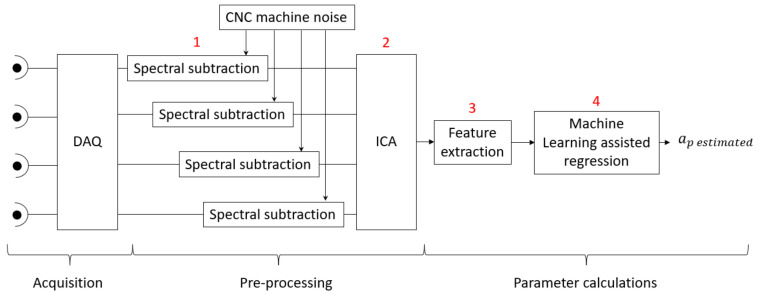
Data processing scheme for the depth of cut calculation $\left(a_{p}\right)$.

**Figure 5 sensors-22-03807-f005:**
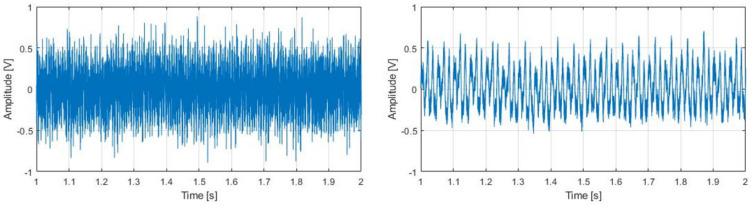
Signal from microphone no.1 introduced in the signal separation algorithm (**left**) and the machining signal resulting from the signal separation of the 4 microphones (**right**).

**Figure 6 sensors-22-03807-f006:**
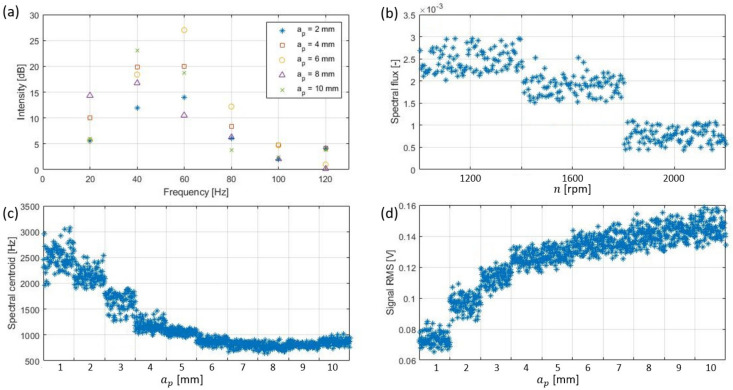
Harmonic intensity of signals with different depths of cut (**a**), spectral flux of signals with different spindle speeds (**b**), spectral centroid of different depths of cut (**c**) and RMS of different depths of cut (**d**).

**Figure 7 sensors-22-03807-f007:**
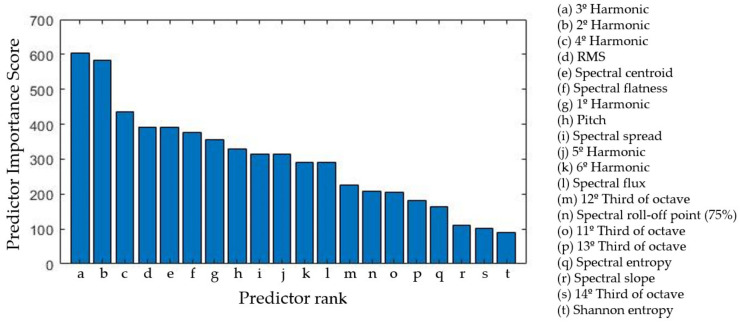
Predictor importance score of each feature differentiating various depths of cut values.

**Figure 8 sensors-22-03807-f008:**
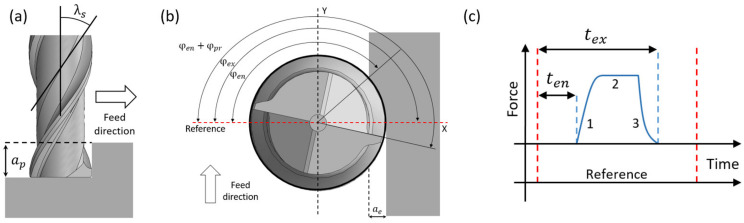
Main parameters of the milling process. (**a**) Side view, (**b**) top view and (**c**) cutting force evolution during the machining process.

**Figure 9 sensors-22-03807-f009:**
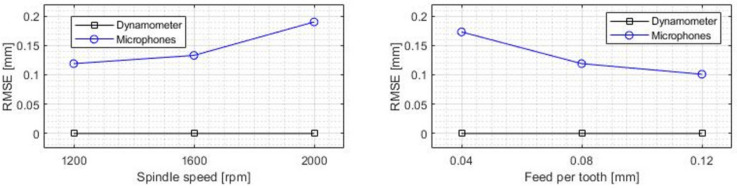
Mean value of the mean square error for each combination of machining parameters.

**Figure 10 sensors-22-03807-f010:**
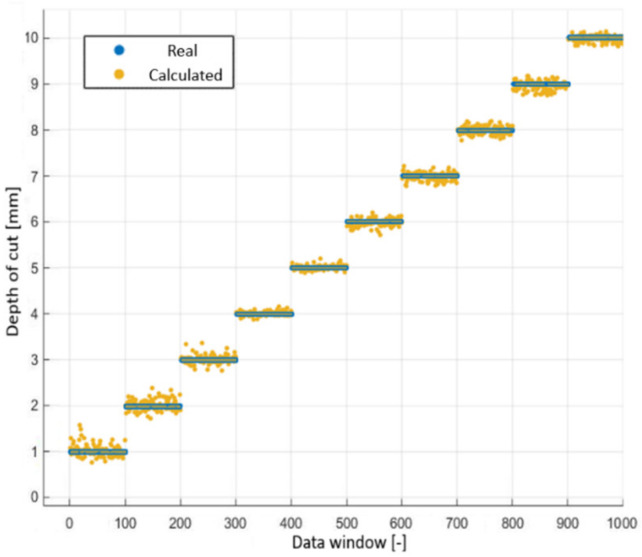
Depth values calculated using the regression model with a feed rate of 0.12 mm at 1200 rpm.

**Figure 11 sensors-22-03807-f011:**
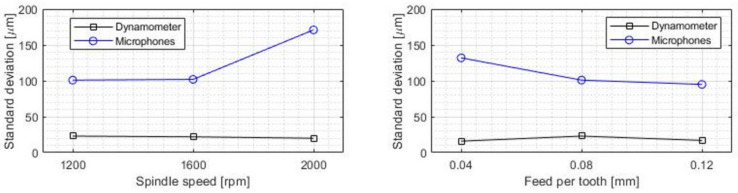
Mean standard deviation for each combination of machining parameters.

**Table 1 sensors-22-03807-t001:** Characteristics of the BSWA MPA-201 microphone.

IEC61672 Standard	Class 1
Range [Hz]	20–20 k
Dynamic range [dBA]	17–146
Inherent noise [dBA]	17
Open circuit sensitivity [mV/Pa]	50

**Table 2 sensors-22-03807-t002:** Characteristics of the Kistler 9257A dynamometer.

Dynamic range [kN]	±5
Threshold [N]	<0.01
Pretensioning direction	Vertical
Frequency range [Hz]	1–2 k
Linearity, all ranges	<±1%

**Table 3 sensors-22-03807-t003:** Machining parameters during the experimental process.

Depth of cut	$a_{p}$ [mm]	1–10
Width of cut	$a_{e}$ [mm]	1
Feed per tooth	$f_{z}$ [mm]	0.04–0.08–0.12
Spindle speed	n [rpm]	1200–1600–2000
Tool flute number	N [-]	1
Tool diameter	*D* [mm]	8
Tool helix angle	$\lambda_{s}\;$[°]	30

**Table 4 sensors-22-03807-t004:** Depth values calculated using the regression model with a feed rate of 0.12 mm at 1200 rpm.

Nominal Depth of Cut [mm]	Feed [mm]	n [rpm]	Dynamometer Average[mm]	Dynamometer STD [µm]	Microphone Average [mm]	Microphone STD [µm]	Absolute Microphone Error [mm]	Relative Microphone Error [%]
1	0.12	1200	1.01	8	1.03	139	0.02	−1.98
2			2.01	9	2.02	125	0.01	−0.50
3			3.00	12	3.01	94	0.01	−0.33
4			3.98	14	4.01	45	0.03	−0.75
5			4.98	17	5.01	51	0.03	−0.60
6			5.99	18	5.99	85	0.00	0.00
7			6.98	17	6.99	91	0.01	−0.14
8			8.02	22	7.99	100	−0.03	0.37
9			9.01	24	8.97	120	−0.04	0.44
10			10.02	32	9.97	101	−0.05	0.50

## Data Availability

The data presented in this study are available on request from the corresponding author.

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
