# Peer review of "Improved Estimation of End-Milling Parameters from Acoustic Emission Signals Using a Microphone Array Assisted by AI Modelling"

_sensors, 2022, doi:10.3390/s22103807_

Round 1
Reviewer 1 Report
The submitted manuscript is a paper that does not particularly shine for its novelty, or complexity of work done, but it’s fair and written. The paper turns out to be easily readable and complete. In the introduction the state of the art of the topic is clearly summarized, and the structure of the paper is effectively outlined. My suggestions are just minor and dealing with few aspects to improve in order to gain more readability and completeness.
At the end of the introduction, when the following parts of the paper are presented, the introduction is mentioned as section 1. It could be avoided, since the introduction has been already read and it could be more intuitive to start listing from section 2.
Section 2,3 and 5 are exhaustive, while section 4 need improvements. Section 6 is suitable for carrying out the task of summarizing the findings of the work.
In section 3, it could be useful to cite more references, in order to understand why the various choices are made that way and where other examples of those methods can be found in literature.
Author Response
Dear reviewer
We sincerely appreciate your suggestions as they have help us to improve our manuscript. Please, check the revised version with the following improvements:
- At the end of the introduction, when the following parts of the paper are presented, the introduction is mentioned as section 1. It could be avoided, since the introduction has been already read and it could be more intuitive to start listing from section 2.
Description of the section at the end of the introduction has been removed to avoid redundancy.
- Section 2,3 and 5 are exhaustive, while section 4 need improvements. Section 6 is suitable for carrying out the task of summarizing the findings of the work.
Section 4 (Validation) has been extended to describe in more detail the fundamentals of the validation method with additional insights about the geometric model referenced in the text and the procedure used to validate our results.
- In section 3, it could be useful to cite more references, in order to understand why the various choices are made that way and where other examples of those methods can be found in literature.
Additional references of other examples in literature have been added regarding the spectral subtraction [35,36], the hypothesis of background noise + signal [38] and the presence of harmonic peaks in the AE signal [40].
We hope that we have addressed all suggestions correctly and we remain at your disposal for any further comments.
Best regards.
A. Sio, on behalf of the authors.
Reviewer 2 Report
The manuscript presents an interesting Improved estimation technique for end-milling parameters from acoustic emission signals.
First of all, I suggest to make even more clear that the term Acoustic emission is referred to the range of audible sound and not to ultrasound, like in case of other monitoring techniques.
In addition, the authors should explain, at least phenomenologically, the physics that should relate the acquired sound with the physical process of milling (e.g. the amplitude is related to the milling depth? the frequency is related to the tool rotational speed? and so on).
Apparently, in the manuscript everything is left to the AI and machine learning tools, and this (in my opinion) is not completely satisfactory.
Author Response
Dear reviewer
We sincerely appreciate your suggestions as they have help us to improve our manuscript. Please, check the revised version with the following improvements:
- First of all, I suggest to make even more clear that the term Acoustic emission is referred to the range of audible sound and not to ultrasound, like in case of other monitoring techniques.
Additional comments specifying that this work is based on analyzing the AE signal in the audible range have been added in the Abstract (lines 18) and Introduction (46 and 78) sections.
- In addition, the authors should explain, at least phenomenologically, the physics that should relate the acquired sound with the physical process of milling (e.g. the amplitude is related to the milling depth? the frequency is related to the tool rotational speed? and so on).
Section 3.2 (Feature extraction) has been expanded to explain the physical influence of the milling parameters in each feature, as this has been the main reason to use these features and not others as opposed to “blind feature extraction” of black-box algorithms which make it more difficult to interpret and learn anything from the results obtained.
- Apparently, in the manuscript everything is left to the AI and machine learning tools, and this (in my opinion) is not completely satisfactory.
We completely agree AI algorithms must be properly used to improve not only the performance of measurement systems but also our knowledge, and therefore special care must be taken to choose the best suited algorithms for each case and to assess the explainability of the developed model. This is the main reason why in this work we decided to use a regression model based on a set of pre-defined features instead of other black box alternatives (the other being related to its simplicity towards a future real time implementation “in the sensor”).
In this work AI comes in last, once we have filtered and combined our signals and extracted a set of features from them which have been derived from the underlying physics of the process. In this last stage is where ML comes in as an effective mean to obtain a non-linear regression
The selection of the different features was made according to the current knowledge of the milling processes and how they are influenced by changes in the machining parameters. Additionally, the influence of each feature was measured to improve the explainability of the system versus the “black box” algorithms like neural networks, whose inner ponderations have a complex interpretability.
We have tried to emphasized this in Section 6 (Conclusions).
We hope that we have addressed all suggestions correctly and we remain at your disposal for any further comments.
Best regards.
A. Sio, on behalf of the authors.
Reviewer 3 Report
The authors have proposed a data-driven approach to estimate the depth of the cut during a milling operation by analyzing the acoustic emission signals with the aid of machine learning. The paper was well written and thoroughly investigated. For clarity of the manuscript, this reviewer has the following comments:
- Line 78: Please recheck the "allow" verb form.
- Please recheck the range (Hz) shown in Table 2.1
- Please recheck the typing of the latent variables (see Line 271).
- Please check the fractional parts of numbers listed in tables 5.1 and 2.2.
- Please recheck the sentence (see Lines 291-292).
The table number is not in consecutive order in this manuscript.
Author Response
Dear reviewer
We sincerely appreciate your suggestions as they have help us to improve our manuscript. Please, check the revised version with the following improvements:
- Line 78: Please recheck the "allow" verb form.
Changed to the “allows” form.
- Please recheck the range (Hz) shown in Table 2.1
The range has been checked.
- Please recheck the typing of the latent variables (see Line 271).
The typing of the latent variables of line 270 has been checked.
- Please check the fractional parts of numbers listed in tables 5.1 and 2.2.
The decimal separator is now a period in every instance.
- Please recheck the sentence (see Lines 291-292).
The paragraph related to the former Equation 9 (now Equation 10) has been modified (lines 293-295).
- The table number is not in consecutive order in this manuscript.
In section 5 (Results and discussion), there was a table erroneously marked as “2.2”. This table is now marked as “5.2”.
We hope that we have addressed all suggestions correctly and we remain at your disposal for any further comments.
Best regards.
A. Sio, on behalf of the authors.